# BiTrajDiff: Bidirectional Trajectory Generation with Diffusion Models for Offline Reinforcement Learning

## Abstract

Recent advances in offline Reinforcement Learning (RL) have proven that effective policy learning can benefit from imposing conservative constraints on pre-collected datasets. However, such static datasets often exhibit distribution bias, resulting in limited generalizability. To address this limitation, a straightforward solution is data augmentation (DA), which leverages generative models to enrich data distribution. Despite the promising results, current DA techniques focus solely on reconstructing *future trajectories* from given states, while ignoring the exploration of *history transitions* that reach them. This single-direction paradigm inevitably hinders the discovery of diverse behavior patterns, especially those leading to critical states that may have yielded high-reward outcomes. In this work, we introduce Bidirectional Trajectory Diffusion (BiTrajDiff), a novel DA framework for offline RL that models both future and history trajectories from any intermediate states. Specifically, we decompose the trajectory generation task into two independent yet complementary diffusion processes: one generating forward trajectories to predict future dynamics, and the other generating backward trajectories to trace essential history transitions. BiTrajDiff can efficiently leverage critical states as anchors to expand into potentially valuable yet underexplored regions of the state space, thereby facilitating dataset diversity. Extensive experiments on the D4RL benchmark suite demonstrate that BiTrajDiff achieves superior performance compared to other advanced DA methods across various offline RL backbones.

## 1 Introduction

Offline Reinforcement Learning (Offline RL) (Fujimoto et al., 2019) seeks to derive effective decision policies exclusively from pre-collected datasets, obviating the necessity for online exploration due to the inherent risks and costs associated with real-time interaction (Levine et al., 2020). This paradigm has demonstrated notable efficacy across various real-world applications, including robotic control (Qing et al., 2024; 2022; Liu et al., 2023) and power grid management (Chen et al., 2024; Xu et al., 2024). A primary challenge in offline RL is addressing the out-of-distribution (OOD) problem, stemming from the inherent divergence between the static offline dataset distribution and the learned policy behavioral distribution during training. Consequently, the direct application of conventional online RL methods inevitably leads to extrapolation error (Levine et al., 2020), characterized by the overestimation of value functions for OOD state-action pairs. To tackle the OOD problem, offline RL research emphasizes the development of techniques that constrain learned policies to remain within the distribution of the static offline dataset. These approaches include imposing explicit constraint terms during policy optimization (Qing et al., 2024; Chen et al., 2022) and penalizing the value function estimates for OOD actions (Lyu et al., 2022; Xu et al., 2023).

Despite promising advancements in offline RL, current algorithms critically restrict agent actions, as evidenced by (Levine et al., 2020; Fujimoto et al., 2019; Qing et al., 2024). This limitation stems from the inherent mechanism constraining learned policy to the biased distribution of offline datasets, often leading to overfitting on limited behavioral patterns. While this principle effectively facilitates the replication of favorable behavioral patterns within the data, it inevitably curtails generalizability, hindering the exploration of optimal behaviors beyond dataset coverage. A common approach to mitigate the distribution bias is Data Augmentation (DA) method, which enriches the offline dataset

with a large amount of synthesized high-fidelity data. For instance, studies like (Lu et al., 2023; Li et al., 2024; Li and Zhang, 2024) leverage one generative diffusion model to approximate the distribution of single-direction forward-future trajectories for augmenting offline data. Notably, as data augmentation (DA) methods operate by modifying the training dataset, they are orthogonal to specific offline RL algorithms, enabling them to enhance any such algorithm without altering its core optimization mechanisms. However, a critical limitation of DA methods is their exclusive focus on trajectories originating from a predefined initial state. While these generated single-direction trajectories effectively explore behaviors emerging from a given state, they neglect the exploration of trajectories leading to that state. This oversight restricts the utility of the generated trajectory to locally explore the trajectory space immediately surrounding existing data, specifically for transitions originating from states already in the dataset. Consequently, while DA methods successfully generate trajectories for local state pairs already reachable within the offline dataset, they are ineffective in discovering global trajectories for previously unreachable state pairs. This naturally leads us to a neglected yet unresolved question:

> ***Given an intermediate state, can we not only imagine its future trajectory but also reconstruct its possible history trajectory leading to the state?***

In this paper, we propose the Bidirectional Trajectory Diffusion framework, abbreviated as BiTrajDiff, a novel data augmentation (DA) approach designed to generate globally diverse behavioral patterns beyond those present in offline datasets. Diverging from conventional DA approaches **generating only local, single-direction trajectories starting from states in the dataset**, BiTrajDiff synthesizes bidirectional trajectories by **concurrent generation of forward-future and backward-history trajectories, conditioned on intermediate anchor states**. This paradigm enables global connectivity between originally unreachable states. Technically, we decouple the bidirectional trajectory generation with two diffusion models: a forward trajectory diffusion and a backward trajectory diffusion. The forward model, conditioned on an initial state, generates subsequent state sequences, while the backward model, conditioned on a terminal state, generates preceding state sequences. This design enables us to stitch these two types of generated trajectories by conditioning them on the same intermediate states. To ensure high-quality and complete generated data, a supervised learning-based inverse dynamics model and a reward model are concurrently deployed to imbue the state sequences with action and reward signals, followed by a novel filtering mechanism that excludes OOD and suboptimal trajectories. Our core idea is that synchronously imagining the forward-future and backward-histroy trajectory allows independent exploration of potential transitions from both temporal directions, and their intersection at learned intermediate states creates novel paths that logically "stitch" these separately explored regions, thereby globally connecting states with no direct transitional evidence in the original data. Therefore, augmenting offline RL with BiTrajDiff-generated data can lead to substantial performance improvements.

**Our contributions** are summarized as follows: **(1)** We are the first to explore and address the challenge of concurrently imagining both future and historical trajectories on a given state for data augmentation in offline RL. This paradigm enables the acquisition of diverse behavioral patterns. **(2)** We propose BiTrajDiff, a novel framework that generates global trajectories between previously unreachable states in the original dataset, achieved by independently generating and then stitching together both forward-future and backward-history trajectories conditioned on common intermediate states. **(3)** We conduct extensive experiments on the D4RL benchmark, applying BiTrajDiff across various offline RL algorithms. Our results demonstrate that BiTrajDiff achieves significantly superior performance compared to other advanced DA baselines, highlighting its effectiveness and robustness.

## 2 RELATED WORKS

**Offline Reinforcement Learning** encompasses four primary categories: policy constraint (Wang et al., 2022; Xu et al., 2023), value regularization (Kostrikov et al., 2021; Kumar et al., 2020), model-based (Yu et al., 2020; Chemingui et al., 2024), and return-conditioned supervised learning (Chen et al., 2021; Paster et al., 2022). Policy constraint methods restrict policies to the offline dataset, employing techniques such as explicit behavior cloning regularization (Qing et al., 2024; Chen et al., 2022) or implicit optimization towards in-sample optimal policies (Nair et al., 2020; Yue et al., 2022). Meanwhile, value regularization promotes conservative value functions to alleviate OOD

overestimation via establishing precise lower bounds for Q-values to facilitate in-sample action selection (Lyu et al., 2022; Park et al., 2024). Model-based approaches construct task-oriented world models from offline data, including one-step dynamics (Yu et al., 2020; Chemingui et al., 2024) and multi-step sequence models (Hafner et al., 2020; 2023), which are subsequently used for offline reward penalization (Yu et al., 2021; Kidambi et al., 2020) and online decision planning to ensure in-distribution actions (Hansen et al., 2022; 2023). Return-conditioned supervised learning trains trajectory generators conditioned on returns, with transformers (Chen et al., 2021; Schmied et al., 2024) and diffusion models (Janner et al., 2022; Ajay et al., 2022) as common backbones. Despite their effectiveness, these approaches are fundamentally limited by reliance on the dataset distribution.

**Data Augmentation for Offline RL** synthesizes additional interactions to enrich the dataset for subsequent RL training. Early work employs dynamics ensembles to generate transitions with uncertainty weighting (Zhang et al., 2023) or diffusion models for flexible one-step oversampling (Lu et al., 2023). More recent efforts shift toward trajectory-level generation. Some work focuses on on-policy rollouts, guided either by policy outputs (Jackson et al., 2024) or interaction histories (He et al., 2023). Other approaches explore prioritized augmentation based on dynamics error or curiosity (Wang et al., 2024), as well as trajectory recomposition through reverse generation (Yang and Wang, 2025) or diffusion-based stitching (Li et al., 2024). Despite these advances, existing DA methods remain limited by their forward-only generation paradigm, restricting exploration to local variations around the dataset. Our BiTrajDiff overcomes this limitation by jointly generating forward and backward trajectories from intermediate states.

## 3 PRELIMINARY

**Reinforcement Learning** paradigm (Sutton and Barto, 2018) is formally defined as a Markov Decision Process (MDP), $\mathcal{M} = \langle \mathcal{S}, \mathcal{A}, P, r, \gamma, \rho_0 \rangle$, where $\mathcal{S}$ is the state space, $\mathcal{A}$ is the action space, $P : \mathcal{S} \times \mathcal{A} \times \mathcal{S} \to [0, 1]$ represents the environment dynamics, $r : \mathcal{S} \times \mathcal{A} \to \mathbb{R}$ is the reward function, $\gamma \in [0, 1)$ is the discount factor, and $\rho_0$ denotes the initial state distribution. At each timestep $t$, an agent executes an action $a_t$ according to its policy $\pi(\cdot|s_t)$ in the current state $s_t$, transitioning to the subsequent state $s_{t+1}$ with reward $r_t$ based on $P$. The objective of the agent is to find an optimal policy $\pi^*$ that maximizes the expected discounted return: $\pi^* = \arg\max_\pi \mathcal{J}(\pi) = \mathbb{E}[\sum_{t=0}^\infty \gamma^t r_t]$. In contrast to online RL, offline RL (Levine et al., 2020) relys solely on a static trajectories dataset $\mathcal{D} = \{\tau^i\}_{i=1}^N$, where $N$ is the number of trajectories and each trajectory $\tau^i = \{(s_t^i, a_t^i, r_t^i)\}_{t=0}^{T-1}$ comprises a sequence of state-action-reward tuples.

**Diffusion Models** (Ho et al., 2020; Song et al., 2021) are generative frameworks that learn data distributions through a forward diffusion process and a reverse denoising process. The forward process gradually corrupts data $\mathbf{x}_0$ via a predefined noise schedule $\{\beta_k\}_{k=1}^K$, where each transition is defined as $q(\mathbf{x}_k|\mathbf{x}_{k-1}) = \mathcal{N}\left(\mathbf{x}_k; \sqrt{1-\beta_k}\mathbf{x}_{k-1}, \beta_k\mathbf{I}\right)$. As $K \to \infty$, the terminal distribution $q(\mathbf{x}_K)$ converges to an isotropic Gaussian $\mathcal{N}(\mathbf{0}, \mathbf{I})$. The reverse process, parameterized by $\theta$, aims to iteratively reconstruct the original data through learnable Gaussian transitions: $p_\theta(\mathbf{x}_{k-1}|\mathbf{x}_k) = \mathcal{N}\left(\mathbf{x}_{k-1}; \mu_\theta(\mathbf{x}_k, k), \Sigma_\theta(\mathbf{x}_k, k)\right)$. By maximizing the Empirical Lower Bound (ELBO) (Sohn et al., 2015) on the log-likelihood of the sampled data, the diffusion model can be trained with a simplified surrogate loss (Sohn et al., 2015):

$$\mathcal{L}_{\text{denoise}}(\theta) = \mathbb{E}_{\mathbf{x}_0 \sim q, k \sim \mathcal{U}\{1,K\}, \epsilon \sim \mathcal{N}(\mathbf{0},\mathbf{I})} \left[ \|\epsilon_\theta(\mathbf{x}_k, k) - \epsilon\|_2^2 \right], \tag{1}$$

where $\mathcal{U}$ denotes the discrete uniform distribution, and $\epsilon_\theta$ is the deep model parameterized with $\theta$ to predict the noise. Both the mean $\mu_\theta(\mathbf{x}_k, k)$ and covariance $\Sigma_\theta(\mathbf{x}_k, k)$ of the reverse process are analytically derivable from $\epsilon_\theta$. The sampling is then performed by initializing with $\mathbf{x}_K \sim \mathcal{N}(\mathbf{0}, \mathbf{I})$ and iteratively applying the learned reverse transitions.

## 4 METHODOLOGY

This section introduces the BiTrajDiff framework for offline RL, which comprises two core components: *bidirectional diffusion training* and *bidirectional trajectory generation*. During the *bidirectional diffusion training* phase, two distinct diffusion models are employed to model the distributions of forward-future and backward-history state sequences within trajectory data. Conditioned on an

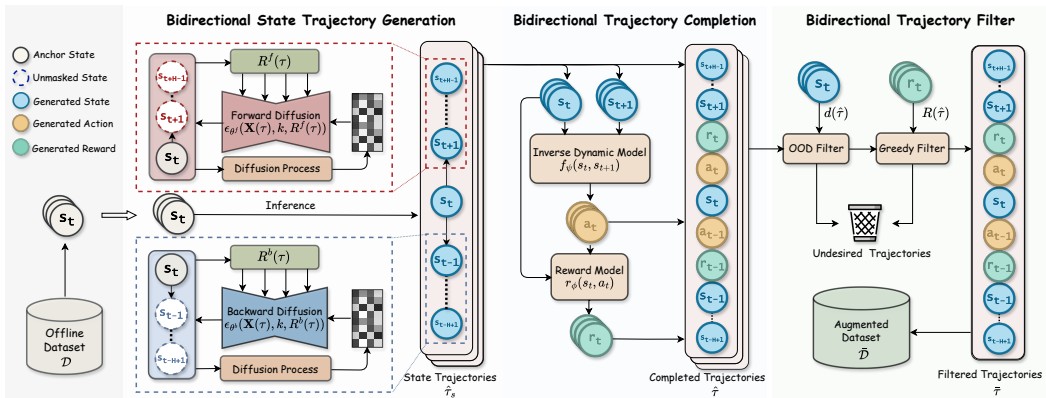

Figure 1: An illustrative diagram of our *Bi*directional *Traj*ectory *Diff*usion (BiTrajDiff) method.

intermediate state and cumulative reward signals, these models generate state sequences in corresponding temporal directions. Then, in the subsequent *bidirectional trajectory generation* phase, the generated forward and backward trajectories are reconciled through filling and filtering operations to produce novel global trajectories that connect previously unreachable states in the original dataset. The framework of our method is presented in Figure 1.

### 4.1 BIDIRECTIONAL DIFFUSION TRAINING

Building upon recent advances in diffusion-based sequence generation (Fathi et al., 2025; Gong et al., 2022), we propose a bidirectional trajectory generation framework comprising independent forward and backward trajectory diffusion models. These models are separately trained on offline datasets to learn the trajectory distribution of the behavior policy, facilitating bidirectional state trajectory generation from shared intermediate "anchor" states.

Formally, for each diffusion model, the state trajectory is generated simultaneously per time step $t$ over the planning horizon $H$: $\mathbf{x}_k(\tau) = \{s_t, s_{t+1}, \cdots, s_{t+H-1}\}_k$, where $k$ denotes the denoise timestep and $t$ represents the MDP timestep. In this way, the conditional generative problem via the diffusion model can be formulated as:

$$\min_\theta \mathcal{L}(\theta) = \mathbb{E}_{\tau \sim \mathcal{D}}[\log p_\theta(\mathbf{x}_0(\tau)|\mathbf{y}(\tau))], \tag{2}$$

where $\mathbf{x}_0(\tau)$ denotes the final generated subtrajectory of diffusion, $\mathbf{y}(\tau)$ represents the condition for generation, and $\theta$ represents the learnable parameters of the diffusion model. While various condition metrics are available, such as temporal difference error, Q-function (Wang et al., 2024), our framework utilizes the cumulative return $R$ and the state $s$ as conditional input $\mathbf{y}(\tau)$. This approach, consistent with prior works (Ajay et al., 2022; Ding et al., 2024), enables the generation of trajectories with desired returns starting from the specific states. However, to enable bidirectional trajectory generation, the formations of the backward and forward diffusion condition inputs are quite distinct.

Specifically, the forward diffusion model generates finite-horizon state trajectories $\mathbf{x}_0^f(\tau) = \{s_t, s_{t+1}, \cdots, s_{t+H-1}\}$ **started at initial state** $s_t$ **with target return** $R^f(\tau) = \sum_{i=t}^{t+H} \gamma^{i-t} r_i$, thus generating physically consistent future trajectories within the offline distribution. Conversely, the backward diffusion model reconstructs historical trajectories $\mathbf{x}_0^b(\tau) = \{s_{t-H+1}, \cdots, s_t\}$ that **ends on terminal state** $s_t$ **and accrued return** $R^b(\tau) = \sum_{i=t-H}^{t} \gamma^{t-i} r_i$. For brevity and clarity, we utilize superscript-free notation (e.g., $\mathbf{x}_k, \mathbf{y}, R(\tau)$) in the following sections to denote computational processes common to both directional models.

Based on the two distinct conditions, we can model the conditional trajectory generation processes by the diffusion sampling composed of the forward noising process $q(\mathbf{x}_{k+1}(\tau)|\mathbf{x}_k(\tau))$ and the reverse denoising process $p_\theta(\mathbf{x}_{k-1}(\tau)|\mathbf{x}_k(\tau), \mathbf{y}(\tau))$ constructed with the diffusion denoise model $\epsilon_\theta$. Meanwhile, the forward and backward diffusion models $\epsilon_\theta$ in our BiTrajDiff framework are parameterized with $\theta^b$ and $\theta^f$, respectively. Thus, given the condition $\mathbf{y}(\tau)$, they are able to generate

the denoised trajectory $\mathbf{x}_0(\tau)$. This process begins with sampling random noise $\mathbf{x}_K \sim \mathcal{N}(\mathbf{0}, \mathbf{I})$, followed by iterative denoising steps where $\mathbf{x}_{k-1} \sim p_\theta$, for $k = \{K, \cdots, 1\}$.

To realize conditional diffusion model training and inference, we employ the Classifier-Free Guidance (CFG) technique (Ho and Salimans, 2022), which uses a single denoise model to handle both conditional and unconditional generation. Formally, the perturbed noise after CFG can be formed as:

$$\hat{\epsilon} = \omega * \epsilon_\theta(\mathbf{x}_k(\tau), k, \mathbf{y}(\tau)) + (1 - \omega) * \epsilon_\theta(\mathbf{x}_k(\tau), k, \emptyset), \qquad (3)$$

where the scalar $\omega \in [0, 1]$ denotes the guidance weight of CFG. Setting $\omega$ to 0 degrades the generation to unconditional generation. Conversely, an increased value of $\omega$ promotes a stronger adherence of the generated samples to the provided condition $\mathbf{y}(\tau)$. To train such diffusion denoise models with CFG, the diffusion denoisers are trained with an objective that enables them to predict noise conditioned on $\mathbf{y}(\tau)$ and also unconditionally. This is typically achieved by randomly dropping the condition $\mathbf{y}(\tau)$ during training with probability $p$. Specifically, we have the following loss function:

$$\mathcal{L}_{\text{denoise}}(\theta) = \mathbb{E}_{\tau \sim \mathcal{D}, k \sim \mathcal{U}\{1,K\}, \epsilon \sim \mathcal{N}(\mathbf{0}, \mathbf{I}), \beta \sim \text{Bern}(p)} \left[ \| \epsilon - \epsilon_\theta(\mathbf{x}_k(\tau), k, (1 - \beta)\mathbf{y}(\tau) + \beta\emptyset) \|^2 \right], \quad (4)$$

where $\text{Bern}(p)$ represents the Bernoulli distribution. In our implementation, we fix the state condition $s_t$ in $\mathbf{x}_k(\tau)$ during the adding-noise training as well as the denoising inference stage for simplicity and efficiency following (Ajay et al., 2022; Yang and Wang, 2025), while the diffusion denoise models $\epsilon_\theta$ only take the cumulative return $R(\tau)$ components of $\mathbf{y}(\tau)$ as condition inputs.

## 4.2 BIDIRECTIONAL TRAJECTORY GENERATION

Based on the pretrained bidirectional diffusion model, we introduce a novel data-augmentation pipeline through three key stages: bidirectional state trajectory generation, bidirectional trajectory completion, and bidirectional trajectory filtering. The pipeline synthesizes global trajectories, thereby establishing connections between states not accessible in the original datasets. This process extends behavioral coverage while upholding physical consistency, ultimately producing augmented data of requisite quality for integration with offline RL algorithms.

### 4.2.1 BIDIRECTIONAL STATE TRAJECTORY GENERATION

We utilize the pretrained forward and backward trajectory diffusion model in section 4.1 to generate bidirectional state trajectories. Prior methods (Lu et al., 2023; Li and Zhang, 2024) model only the conditional distribution of trajectories originating from a given state, neglecting the historical pathways leading to those states. As a result, these data augmentation techniques yield only limited expansion in behavior diversity. To address this limitation without introducing additional noise, our proposed method, BiTrajDiff, generates both forward and backward trajectories and integrates them by stitching at a shared intermediate state $s_t$, which serves as a consistent anchor.

Specifically, we sample the candidate state $s$ from the offline dataset $\mathcal{D}$ as the condition state $s_t$. By combining the state $s_t$ with the manually selected return signal $R(\tau)$ into the $\mathbf{y}(\tau)$, we can leverage the pre-trained bidirectional diffusion model $\theta$ to generate two distinct trajectories: a forward-future trajectory $\mathbf{x}_0^f(\tau)$ originates from $s_t$, and a backward-history trajectory $\mathbf{x}_0^b(\tau)$ concludes at $s_t$. Since $s_t$ is both the start state of $\mathbf{x}_0^f(\tau)$ and the end state of $\mathbf{x}_0^b(\tau)$, we can directly stitch them to construct the bidirectional state trajectory $\hat{\tau}_s(s_t)$. Formally, let $\mathbf{x}_0(\tau)[i:j]$ represents the observation slice from time $i$ to time $j$, the stitched bidirectional state trajectory $\hat{\tau}_s(s_t)$ can be formulated as:

$$\hat{\tau}_s(s_t) = \left\{ \mathbf{x}_0^b(\tau)[0:H-1], s_t, \mathbf{x}_0^f(\tau)[1:H] \right\} = \{\hat{s}_{t-H+1}, \cdots, \hat{s}_{t-1}, s_t, \hat{s}_{t+1}, \cdots, \hat{s}_{t+H-1}\}, \quad (5)$$

where $H$ is the horizon, and $\hat{s}_i$ is the diffusion generated states at MDP timestep $i$. By conditioning on and concatenating at the intermediate anchor state $s_t$, we construct a state trajectory, $\hat{\tau}_s(s_t)$, that connects $\hat{s}_{t-H+1}$ and $\hat{s}_{t+H-1}$. This process enables the connection of states that may be disconnected in the original dataset, thereby substantially enriching the diversity of observed behavioral patterns.

### 4.2.2 BIDIRECTIONAL TRAJECTORY COMPLETION

In this completion process, the generated bidirectional state trajectories $\hat{\tau}_s(s_t)$ are further completed by adding the action and reward signal. We introduce the Inverse Dynamics Model (IDM):

$f_\psi(s_t, s_{t+1}) : \mathcal{S} \times \mathcal{S} \to \mathcal{A}$ and the Reward Model (RM): $r_\phi(s_t, a_t) : \mathcal{S} \times \mathcal{A} \to \mathbb{R}$. The supervised training objective of the two models is:

$$\mathcal{L}(\psi, \phi) = \mathbb{E}_{(s_t, a_t, s_{t+1}) \sim \mathcal{D}} \left[ \| f_\psi(s_t, s_{t+1}) - a_t \|^2 + \| r_\phi(s_t, a_t) - r_t \|^2 \right] \quad (6)$$

Thus, by feeding the generated state trajectory $\hat{\tau}_s(s_t)$ to the trained IDM and RM sequentially, we obtain the completed bidirectional trajectory: $\hat{\tau}(s_t) = \{(\hat{s}_i, \hat{a}_i, \hat{r}_i, \hat{s}_{i+1})\}_{i=t-H+1}^{t+H-2}$ [1], where $\hat{a}_i = f_\psi(\hat{s}_i, \hat{s}_{i+1})$ and $\hat{r}_i = r_\phi(\hat{s}_i, \hat{a}_i)$. Therefore, we can obtain a generated dataset $\hat{\mathcal{D}} = \{\hat{\tau}(s_{t_i})\}_{i=1}^{\hat{N}}$, where $\hat{N}$ is the number of generate trajectories.

### 4.2.3 BIDIRECTIONAL TRAJECTORY FILTER

To ensure dataset quality, we employ a two-stage filtering mechanism that removes both out-of-distribution and suboptimal trajectories. This selective process preserves the reliability of the augmented data without sacrificing diversity. In BiTrajDiff, the trajectory filter comprises an OOD trajectory filter and a greedy trajectory filter.

**OOD Trajectory Filter.** We model the OOD trajectory filter as a novelty detection model (Pimentel et al., 2014), which addresses the identification of data instances that exhibit a significant deviation from the offline training dataset. And we utilize the Isolation Forest model (Liu et al., 2008) for OOD transition detection, which assigns an anomaly score to each data instance, reflecting its susceptibility to isolation with shorter path lengths in randomly partitioned trees. Specifically, we construct the isolation forest on the original dataset $\mathcal{D}$, which is able to give each observation $\hat{s}_i$ an anomaly score $d(\hat{s}_i)$. We defined the OOD score of a generated trajectory $d(\hat{\tau}) = \sum_{i=t-H+1}^{t+H-2} d(\hat{s}_i)$. Then we sort $\hat{\mathcal{D}}$ by $d(\hat{\tau})$ and retain the top-$C_{\text{ood}}$ trajectories with the smallest values, where $C_{\text{ood}}$ is a hyperparameter.

**Greedy Trajectory Filter.** The greedy trajectory filter selects the generated trajectories with the highest cumulative reward. Specifically, we sort the $C_{\text{ood}}$ trajectories retained from OOD trajectory filter by the sum reward $R(\hat{\tau}) = \sum_{i=t-H+1}^{t+H-2} \hat{r}_i$. The top-$C_{\text{greedy}}$ trajectories with the highest sum rewards are picked as the final generated trajectories dataset $\tilde{\mathcal{D}}$ for offline RL training.

In summary, our BiTrajDiff operates in two key stages. Initially, it trains a bidirectional diffusion model to capture the distributions of both forward-future and backward-history trajectories. Subsequently, BiTrajDiff iteratively generates bidirectional trajectory conditioned on intermediate anchor states sampled from $\mathcal{D}$. These generated sequences undergo further completion and filtering to create a high-fidelity augmented dataset $\tilde{\mathcal{D}}$, which is then mixed with $\mathcal{D}$ to train downstream offline RL algorithms. We adopt CleanDiffuser (Dong et al., 2024) as the backbone for BiTrajDiff, owing to its generality and robustness. The pseudocode of BiTrajDiff is presented in Appendix A.

## 5 EXPERIMENTS

To demonstrate the effectiveness of the proposed BiTrajDiff method, we conduct experiments on the D4RL benchmark (Fu et al., 2020). Our evaluation seeks to answer the following questions: (1) Does BiTrajDiff outperform existing DA methods across various offline RL algorithms? (Section 5.2 and Appendix C) (2) How do different components of BiTrajDiff affect the offline RL performance? (Section 5.3 and Appendix D-F) (3) Can BiTrajDiff find the potentially valuable state leading to high return more effectively than other DA baselines? (Section 5.4 and Appendix 5.4) (4) Can BiTrajDiff generate more diverse trajectories compared to single-direction DA approaches? (Section 5.5)

### 5.1 EXPERIMENT SETTINGS

We evaluate our approach on the D4RL benchmark (Fu et al., 2020), focusing on locomotion, navigation, and manipulation tasks. We compare BiTrajDiff with three state-of-the-art DA methods: Synther (Lu et al., 2023),DiffStitch (Li et al., 2024) and RTDiff (Yang and Wang, 2025). To evaluate the effectiveness of them, we conduct extensive experiments on four representative offline RL algorithms: IQL (Kostrikov et al., 2021), TD3BC (Fujimoto and Gu, 2021), CQL (Kumar et al., 2020), and DT (Chen et al., 2021). These methods have been widely adopted as standard baselines in offline RL due to their stable performance. The hyperparameter setting can be viewed in Appendix B.2.

---

[1] For the sake of clarity, the condition state $s_t$ is written as $\hat{s}_t$.

Table 1: Performance of our BiTrajDiff and baselines on the locomotion tasks. ± corresponds to the standard deviation of the performance on 5 random seeds. The best and the second-best results of each setting are marked as **bold** and underline, respectively. Detailed reports about CQL and DT are reported in Appendix C.

| Source | Task | IQL (Kostrikov et al., 2021) | | | | | TD3BC (Fujimoto and Gu, 2021) | | | | |
|---|---|---|---|---|---|---|---|---|---|---|---|
| | | Base | RTDiff | Synther | DiffStitch | Ours | Base | RTDiff | Synther | DiffStitch | Ours |
| medium | halfcheetah | 48.2±0.2 | 48.9±0.2 | **49.2**±0.2 | 48.6±0.2 | 48.6±0.2 | 48.4±0.2 | 49.4±0.2 | 49.7±0.4 | 50.1±0.6 | **50.3**±0.4 |
| | hopper | 67.0±1.9 | 62.0±2.4 | 61.1±2.7 | 65.5±4.7 | **81.1**±5.0 | 60.4±3.7 | 62.6±5.1 | 61.2±6.1 | 67.9±3.7 | **79.0**±3.5 |
| | walker2d | 77.6±5.3 | 82.7±4.0 | 85.8±0.7 | 79.8±1.8 | **86.7**±1.7 | 82.0±2.7 | 85.4±1.1 | 83.4±2.0 | 85.5±0.7 | **86.7**±1.0 |
| medium expert | halfcheetah | 90.1±4.4 | 92.0±3.1 | 86.6±6.2 | 90.3±8.0 | **95.3**±0.1 | 92.3±5.1 | 93.9±2.1 | 95.5±0.8 | 95.9±0.7 | **96.3**±0.4 |
| | hopper | 105.4±6.3 | 110.5±0.5 | **111.2**±0.7 | 108.6±1.2 | 110.9±0.6 | 94.2±11.1 | 106.8±7.0 | 106.2±6.7 | 97.5±9.9 | **109.0**±5.1 |
| | walker2d | 112.4±0.4 | 111.9±0.5 | 112.1±0.6 | 111.2±0.3 | **112.5**±0.5 | 109.3±0.2 | 109.6±0.4 | 109.4±0.0 | 109.6±0.5 | **110.2**±0.3 |
| medium replay | halfcheetah | 43.5±0.2 | 43.7±0.8 | **46.2**±0.3 | 43.6±0.1 | 44.1±0.2 | 44.1±0.1 | 44.0±0.3 | 44.5±0.6 | **45.1**±0.5 | 45.0±0.6 |
| | hopper | 94.7±4.0 | 97.7±2.7 | 100.6±0.4 | 96.4±5.6 | **102.8**±0.6 | 64.4±8.9 | 72.1±18.9 | 76.4±3.0 | 71.5±21.0 | **85.0**±10.3 |
| | walker2d | 73.3±7.5 | 74.0±5.2 | 86.7±1.1 | 74.9±8.2 | **87.6**±2.2 | 80.9±6.0 | 83.5±7.4 | 81.1±5.9 | 80.5±5.2 | **89.9**±1.4 |
| **Average** | | 77.5 | 79.0 | 80.8 | 78.7 | **85.5** | 75.1 | 78.6 | 78.6 | 78.2 | **83.5** |

Table 2: Performance of our BiTrajDiff and baselines on the navigation and manipulation tasks.

| Task | IQL (Kostrikov et al., 2021) | | | | | TD3BC (Fujimoto and Gu, 2021) | | | | |
|---|---|---|---|---|---|---|---|---|---|---|
| | Base | RTDiff | Synther | DiffStitch | Ours | Base | RTDiff | Synther | DiffStitch | Ours |
| maze2d-umaze | 56.0±6.1 | 53.1±2.2 | 51.9±4.8 | 52.1±5.9 | **62.3**±2.7 | 38.1±10.2 | 43.1±6.4 | 40.3±10.0 | 42.5±8.3 | **45.3**±3.1 |
| maze2d-medium | 41.2±2.9 | 50.8±9.6 | 66.9±16.9 | 46.4±10.2 | **72.2**±14.8 | 30.2±16.3 | 36.3±9.9 | 34.8±14.2 | 34.7±3.0 | **45.7**±7.6 |
| maze2d-large | 67.9±4.2 | **74.3**±5.2 | 70.9±2.5 | 64.2±6.1 | 71.3±2.3 | 95.3±22.7 | 102.8±8.9 | 100.6±34.3 | 106.1±11.4 | **129.3**±26.0 |
| antmaze-umaze-diverse | 53.0±4.7 | 61.6±5.7 | 55.4±14.2 | 55.4±5.4 | **63.4**±7.9 | 43.2±7.4 | 45.8±4.4 | 45.0±10.3 | **49.4**±4.8 | 47.0±2.1 |
| antmaze-medium-diverse | 71.2±5.0 | 73.2±8.8 | 81.2±6.4 | 69.2±17.2 | **86.2**±5.8 | 0.0±0.0 | 0.0±0.0 | **6.4**±5.9 | 0.0±0.0 | 4.8±1.7 |
| antmaze-large-diverse | 18.0±8.4 | 53.2±9.5 | 57.8±6.4 | 12.8±15.2 | **63.0**±9.3 | 0.0±0.0 | 0.0±0.0 | 0.4±0.5 | 0.0±0.0 | **1.2**±0.9 |
| **Maze Average** | 51.2 | 61.0 | 64.0 | 50.0 | **69.7** | 28.1 | 38.0 | 37.1 | 38.8 | **45.5** |
| kitchen-complete | 43.3±16.0 | 51.9±7.4 | 42.8±16.4 | 38.3±8.9 | **59.1**±12.3 | 0.1±0.1 | 0.8±0.5 | 0.1±0.2 | 0.5±0.7 | **1.5**±1.0 |
| kitchen-partial | 68.0±5.1 | 69.8±2.9 | **70.3**±2.8 | 62.9±9.4 | 69.7±1.8 | 11.8±9.7 | 8.9±7.0 | 10.9±1.8 | **13.0**±9.3 | 13.0±5.9 |
| kitchen-mixed | 60.7±2.5 | **64.4**±6.1 | 62.6±2.7 | 63.4±5.2 | 62.9±4.9 | 8.7±5.1 | 7.7±0.7 | 10.7±5.0 | 6.0±4.7 | **30.9**±12.6 |
| **Kitchen Average** | 57.3 | 62.0 | 58.6 | 54.9 | **63.9** | 6.9 | 5.8 | 7.2 | 6.5 | **15.1** |

## 5.2 MAIN RESULTS

**Results for Locomotion Tasks.** The experimental results on D4RL locomotion tasks are summarized in Table 1. Our proposed BiTrajDiff demonstrates consistent superiority across all dataset configurations. Specifically, BiTrajDiff achieves performance gains of 10.3% on IQL and 11.2% on TD3BC, substantially exceeding improvements reported by alternative methods like DiffStitch. Meanwhile, results on CQL and DT, presented in Appendix C, exhibit a similar phenomenon. These consistent gains across the offline RL algorithms underscore the effectiveness of our BiTrajDiff approach.

**Results for Maze and Franka Kitchen Tasks.** Table 2 reports the experimental results on the D4RL Maze and Franka Kitchen tasks. BiTrajDiff achieves the highest average return under both the IQL and TD3BC algorithms. Since the Maze and Franka Kitchen tasks are sparse-reward settings, which are typically detrimental to RL training, the availability of high-quality data is particularly critical. BiTrajDiff tackles this challenge by generating forward-future and backward-history trajectories conditioned on intermediate anchor states, stitching them into novel trajectories. Consequently, BiTrajDiff enhances data diversity quality, yielding superior and more stable performance.

## 5.3 ABLATION ANALYSIS

**Ablation Study of the Direction of Diffusion Generation.** To confirm the effectiveness of our bidirectional generation paradigm, we compare the performance improvement with the augmented diffusion trajectories generated in a single direction and in a bidirectional way, as shown in Figure 2. While the forward method yields moderate improvements, it is consistently outperformed because it neglects the historical transitions leading to critical, high-reward states. Conversely, the backward method achieves substantial gains by exploring trajectories toward a predefined initial state, thus mitigating the overestimation risk associated with unknown states (Yang and Wang, 2025). Finally, demonstrating consistently superior and robust performance, our bi-directional framework, BiTrajDiff, empirically validates that by simultaneously synthesizing forward-future and backward-history trajectories, BiTrajDiff constructs novel, high-quality paths absent from the original dataset.

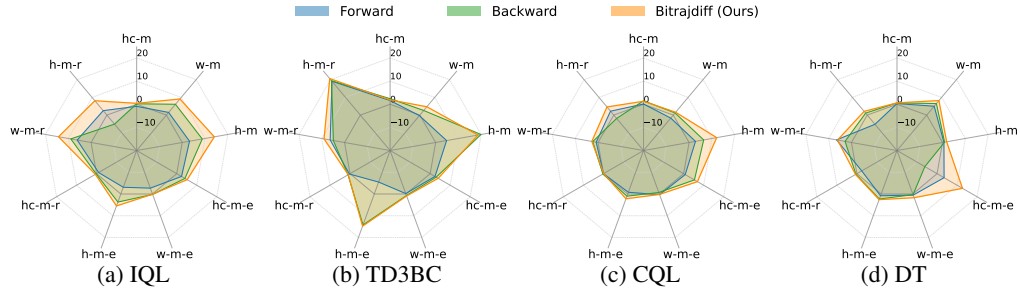

Figure 2: Performance improvement comparison of offline RL algorithms augmented with single- and bi-directional diffusion trajectories. The task abbreviations are listed in Table S1.

**Ablation Study of Trajectory Filters.** We compare the IQL and TD3BC training process augmented by bitrajdiff datasets with different trajectory filter strategies. The learning curves are presented in Figure 3. Without any filters, the training process is highly volatile, as unfiltered, potentially OOD trajectories introduce suboptimal or even illegal states and cause erratic policy updates. In contrast, applying the OOD filter substantially stabilizes the training process by constraining synthetic data to a plausible behavioral space, yet this configuration consistently underperforms the full BiTrajDiff model, which combines both OOD and greedy filters. This is because the greedy filter selectively retains trajectories that yield higher returns, thereby actively guiding the policy toward behaviors with high return and facilitating the discovery of better paths. These findings provide strong evidence for the necessity of employing both OOD and greedy filters.

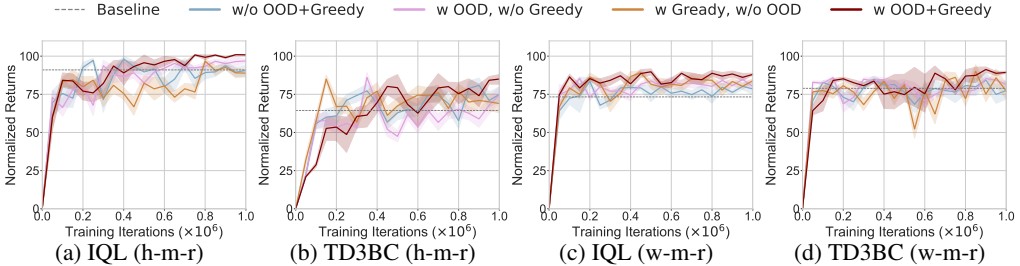

Figure 3: Learning curves of BiTrajDiff with data from different variants related to trajectory filters. Detailed results are reported in Appendix E.

**Ablation Study of Augmented Data Ratio.** We investigate the balance between original and augmented data by varying the ratio $\sigma$ of augmented to original data, with the results presented in Figure 4. When $\sigma$ is too low, performance improvements remain limited across the tested algorithms, illustrating that a limited amount of augmented data is not enough to provide offline RL algorithms with new behavior patterns. Conversely, increasing the ratio to a range of $30\% \sim 50\%$ leads to a substantial performance boost and stability. However, when the ratio is increased to $100\%$, the performance becomes highly unstable, which suggests that the excessive noise introduced by an overabundance of synthetic data hinders offline RL algorithms from learning effectively. Finally, we select $\sigma = 30\%$ as the fixed hyperparameter due to its stable performance improvement.

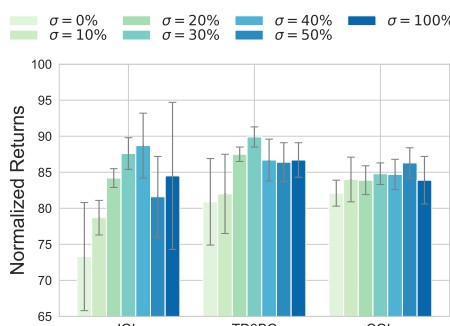

Figure 4: Compare the returns of BiTrajDiff with different augmented data ratios $\sigma$ in the walker2d-medium-replay task. Detailed results are reported in Appendix F.

## 5.4 EFFECTIVENESS UNDER $n$-STEP TD ESTIMATOR

We employ the $n$-step TD estimator to update the critic networks in IQL and TD3BC under the original dataset and the DA-augmented data, and further compare performance across different values of $n$ as shown in Figure 5. The $n$-step TD estimator (De Asis et al., 2018) enables the discovery of high-return solutions by looking ahead from more distant states, making performance improvement a natural indicator of the quality of DA-generated datasets. As shown in Figure 5, all three DA methods consistently outperform the base method, suggesting that the generated trajectories effectively uncover

more rewarding states. However, as $n$ increases, the performance inevitably declines due to the growing variance in value estimation. Meanwhile, for DA methods, this effect is compounded by the accumulated mismatching errors introduced by the inverse dynamics model, which negatively impact TD estimation. In particular, our method consistently achieves the highest performance across all tested settings and maintains its advantage over RTDiff and Diffstitch as the $n$-step horizon increases. These results verify the effectiveness of BiTrajDiff for offline RL algorithms under $n$-step TD estimator. By stitching forward-future and backward-history trajectories, BiTrajDiff obtain a superior capability to synthesize high-quality trajectories compared to existing DA methods.

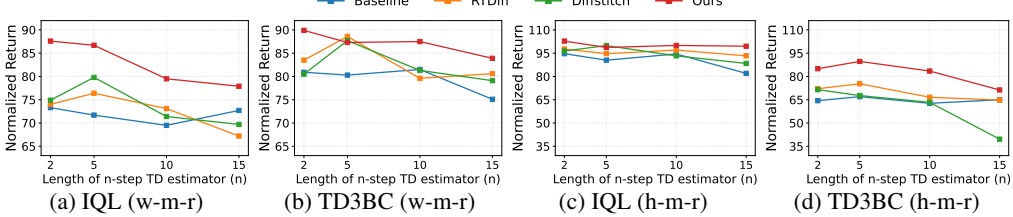

|  |  |  |  |
| :---: | :---: | :---: | :---: |
| (a) IQL (w-m-r) | (b) TD3BC (w-m-r) | (c) IQL (h-m-r) | (d) TD3BC (h-m-r) |

Figure 5: Test returns comparisons between other DA baselines and our BiTrajDiff under varying $n$-step TD estimators. Detailed results are reported in Appendix G.

## 5.5 VISUALIZATION

To evaluate the accuracy and diversity robustness of the BiTrajDiff framework, we compare its generated trajectories with those produced by single-direction forward and backward diffusion models. Following Lu et al. (2023), we employ two quantitative metrics to assess a generated trajectory $\hat{\tau}$: (1) Dynamic Error: $\mathcal{E}_{\text{Dyn}}(\hat{\tau}) = \sum_t \|\hat{s}_{t+1} - s_{t+1}\|_2$, which measures trajectory accuracy by summing the mean squared error between the predicted next states $\hat{s}_{t+1}$ and the ground truth $s_{t+1}$; (2) L2 Distance: $\mathcal{E}_{\text{L2D}}(\hat{\tau}) = \min_{\tau \in \mathcal{D}} \sum_t \|\hat{s}_t - s_t\|_2$, defined as the minimal L2 distance between the generated trajectory and any trajectory of equal length in $\mathcal{D}$, capturing trajectory diversity. As shown in Figure 6, while forward and backward diffusion models achieve low $\mathcal{E}_{\text{Dyn}}$, they exhibit limited $\mathcal{E}_{\text{L2D}}$, indicating that the generated

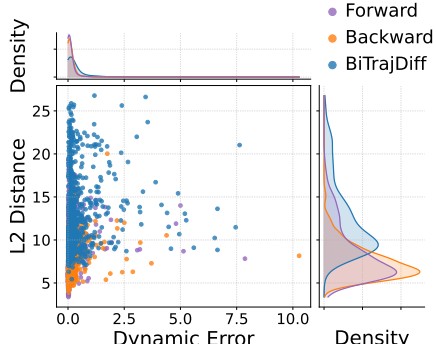

Figure 6: BiTrajDiff vs forward/backward diffusion: dynamic error and L2 distance.

trajectories closely resemble those in the original dataset and thus offer limited performance gains. In contrast, trajectories generated by BiTrajDiff demonstrate significantly broader marginal distributions in $\mathcal{E}_{\text{L2D}}$ while maintaining competitive $\mathcal{E}_{\text{Dyn}}$, reflecting enhanced behavioral diversity without sacrificing dynamic consistency. Consequently, BiTrajDiff facilitates greater performance gains with the augmented data. These results highlight the ability of BiTrajDiff to generate behaviorally diverse trajectories while preserving accuracy, effectively balancing exploration and reliability.

## 6 CONCLUSION

In this paper, we present BiTrajDiff, a novel diffusion-based framework that enhances offline RL by enabling bidirectional trajectory generation from shared intermediate anchor states. Unlike previous single-direction augmentation methods that only model forward future trajectories from observed states, BiTrajDiff simultaneously generates both forward-future and backward-history trajectories. This bidirectional generation strategy allows BiTrajDiff to not only imagine plausible futures but also reconstruct potential historical transitions, thereby creating novel trajectories that globally connect states, including those unreachable in the original dataset. This leads to significantly diverse and effective trajectory augmentation, improving offline RL performance beyond the capability of the forward-only generation schemes of other DA frameworks. Extensive experiments on the D4RL benchmark also demonstrate that BiTrajDiff significantly outperforms existing data augmentation baselines across diverse offline RL algorithms. Future work will explore bidirectional generation for offline RL under multi-task settings with broader behavioral patterns.

## ETHICS STATEMENT

This work does not involve human subjects, potentially harmful insights, potential conflicts of interest and sponsorship, or discrimination concerns, and therefore, we do not foresee any ethical concerns.

## REPRODUCIBILITY STATEMENT

We have provided the code implementation of BiTrajDiff in the supplementary material, while the hyperparameter setting is also provided in Appendix B.2.

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

## A  PSEUDOCODE

The pseudocode of our BiTrajDiff method is provided in Algorithm 1.

---

**Algorithm 1** Bidirectional Trajectory Diffusion (BiTrajDiff) data augmentation pipeline.

---

**Input:** dataset $\mathcal{D}$, ratio $\sigma$, horizon $H$, target return-to-go $R^f$ and $R^b$, filter number $C_{\text{OOD}}$ and $C_{\text{greedy}}$.
**Initialize:** Foward and backward diffusion denoise network $\epsilon_{\theta f}$ and $\epsilon_{\theta b}$, inverse dynamics model $f_\psi$, reward network $r_\phi$, and the augmented dataset $\mathcal{D}_{\text{da}} = \emptyset$
   ▷ BiTrajDiff Model Training
   Train the forward and backward trajectory diffusion model $\epsilon_{\theta f}$ and $\epsilon_{\theta b}$ with $\mathcal{D}$ according to Eq. 4.
   Train the inverse dynamics model $f_\psi$ and reward model $r_\phi$ with $\mathcal{D}$ according to Eq. 6
   ▷ BiTrajDiff Data Generation
   **while** $|\mathcal{D}_{\text{da}}| \leq \sigma * |\mathcal{D}|$ **do**
      # Bidirectional State Trajectory Generation
      Sample random batch of states $\mathcal{B} = \{s_t\} \sim \mathcal{D}$.
      Generate $\mathbf{x}_0^f(\tau) = \{\hat{s}_t, \hat{s}_{t+1}, \cdots, \hat{s}_{t+H-1}\}$ with $\epsilon_{\theta f}$ conditioned on $s_t$ and $R^f$.
      Generate $\mathbf{x}_0^b(\tau) = \{\hat{s}_{t-H+1}, \cdots, \hat{s}_{t-1}, \hat{s}_t\}$ with $\epsilon_{\theta b}$ conditioned on $s_t$ and $R^b$.
      Construct the bidirectional state trajectory $\hat{\tau}_s(s_t)$ with $\mathbf{x}_0^f(\tau)$ and $\mathbf{x}_0^f(\tau)$ according to Eq. 5.
      # Bidirectional Trajectory Completion
      Compute the action $\hat{a}_i$ and reward $\hat{r}_i$ signal with $\hat{\tau}_s(s_t)$ by $f_\psi$ and $r_\phi$.
      Obtain the generated trajectory set $\hat{\mathcal{D}}$ containing $\hat{\tau}(s_t) = \{(\hat{s}_i, \hat{a}_i, \hat{r}_i, \hat{s}_{i+1})\}_{i=t-H+1}^{t+H-2}$
      # Bidirectional Trajectory Filter
      Filter the Top-$C_{\text{ood}}$ trajectories with smalllest $d(\hat{\tau})$ from $\hat{\mathcal{D}}$
      Filter the Top-$C_{\text{greedy}}$ trajectories as batch $\tilde{\mathcal{D}}$ with largest $R(\hat{\tau})$ from the Top-$C_{\text{ood}}$ trajectories.
      Add the final generated batch into the augmented dataset: $\mathcal{D}_{\text{da}} = \mathcal{D}_{\text{da}} \bigcup \tilde{\mathcal{D}}$
   **end while**

---

## B  EXPERIMENTS DETAILS

### B.1  TASK ABBREVIATIONS AND TASK VERSIONS

For improved readability and conciseness, we use abbreviations for the locomotion tasks throughout the main text, with the corresponding definitions provided in Table S1.

### B.2  IMPLEMENTATION DETAILS

#### B.2.1  BITRAJDIFF IMPLEMENTATION

In this section, we provide the implementation details of our experiments. We conducted our experiments on a cluster of 4 A100 GPUs. The source code will be made publicly available upon the publication of this paper. Our BiTrajDiff is implemented based on cleandiffuser (Dong et al., 2024), a recent modularized library for diffusion models in decision making. We represent the noise model $\epsilon_\theta$ with the 1D DiT backbone (Peebles and Xie, 2022), consisting 2 transformer blocks with adaptive layer normalization. Each block contains a multi-head self-attention layer followed by a feed-forward MLP. The diffusion timestep embeddings are first encoded via Fourier features and then projected through a two-layer MLP network with 128 hidden units. Similarly, the condition

Table S1: Abbreviations of the corresponding locomotion tasks and datasets.

| Dataset | halfcheetah | walker2d | hopper |
|---|---|---|---|
| medium | hc-m | w-m | h-m |
| medium-replay | hc-m-r | w-m-r | h-m-r |
| medium-expert | hc-m-e | w-m-e | h-m-e |

Table S2: Test returns of our proposed BiTrajDiff and baselines on the Gym tasks. ± corresponds to the standard deviation of the performance on 5 random seeds. The best and the second-best results of each setting are marked as **bold** and underline, respectively.

| Source | Task | CQL (Kumar et al., 2020) | | | | | DT (Chen et al., 2021) | | | | |
|---|---|---|---|---|---|---|---|---|---|---|---|
| | | Base | RTDiff | Synther | DiffStitch | Ours | Base | RTDiff | Synther | DiffStitch | Ours |
| medium | halfcheetah | 47.4±0.2 | 47.8±1.6 | 48.3±0.4 | 48.0±0.2 | **48.7±0.3** | 42.7±0.5 | 42.6±2.5 | 42.8±0.4 | 42.8±0.4 | **43.3±0.2** |
| | walker2d | 82.6±1.0 | 83.5±1.7 | 83.4±0.5 | 82.4±4.5 | **84.2±0.5** | 69.6±11.0 | 71.6±10.2 | 72.3±9.4 | **77.8±4.0** | **77.8±4.0** |
| | hopper | 68.8±4.1 | 70.9±3.7 | 75.5±8.4 | 69.7±5.8 | **80.9±3.4** | 56.0±1.9 | 56.6±5.1 | **58.4±2.4** | **58.4±2.4** | 57.8±2.2 |
| medium expert | halfcheetah | 87.2±4.6 | 90.0±2.9 | 92.2±3.5 | 91.4±5.0 | **94.1±1.1** | 70.9±12.3 | 72.3±7.0 | 76.9±12.7 | **83.5±6.3** | **83.5±6.3** |
| | walker2d | 110.2±0.6 | 110.1±0.8 | 110.1±0.4 | 109.4±3.6 | **110.4±0.4** | 104.6±6.9 | 101.2±1.8 | 104.0±0.6 | **106.3±6.0** | **106.3±6.0** |
| | hopper | 103.5±8.1 | 102.2±3.2 | 103.2±10.2 | 105.3±7.8 | **105.7±5.8** | 107.5±3.4 | 109.6±1.3 | 108.8±4.8 | **110.1±1.9** | **110.1±1.9** |
| medium replay | halfcheetah | 45.9±0.2 | 45.6±0.4 | **47.4±0.6** | 45.1±0.4 | 45.8±0.3 | 38.8±2.4 | 39.1±4.1 | 38.9±2.6 | **39.5±0.8** | **39.5±0.8** |
| | walker2d | 82.1±1.8 | 84.4±3.2 | **84.8±0.6** | 82.4±1.8 | **84.8±1.5** | 52.1±5.5 | 53.8±2.4 | 54.6±7.5 | **58.2±6.1** | **58.2±6.1** |
| | hopper | 95.7±2.5 | 95.9±7.1 | 96.1±1.2 | 98.2±6.8 | **100.3±1.3** | 67.3±8.9 | **70.2±10.6** | 68.1±19.7 | 69.6±13.1 | 69.6±13.1 |
| **Average** | | 80.4 | 81.3 | 82.2 | 80.3 | **84.9** | 68.6 | 69.5 | 70.7 | 73.1 | **73.7** |

inputs are encoded by a two-layer MLP network with 128 hidden units. For the sampling process, we follow the cleandiffuser reimplementation of Decision Diffuser (Ajay et al., 2022), employing the VP-SDE (Song et al., 2020) formulation with 20 diffusion steps for sampling. Meanwhile, the inverse dynamics and rewards model are both represented by a two-layer MLP network with 512 hidden units, respectively. During BiTrajDiff training, we employ an Adam optimizer with a learning rate of $2e-4$ for all networks, using a batch size of $64$ and performing $1e6$ gradient steps in total. The default trajectory horizon $H$ for both training and generation is set to 5. The ratio $\sigma$ of the augmented dataset is 30%. During our BiTrajDiff data augmentation pipeline, the batch size is set to $512$, while the filter number $C_{\text{ood}}$ is 256 and $C_{\text{greedy}}$ is 64. As for the target return-to-go $R^f$ and $R^b$, we set them to be equal in our BiTrajDiff conditional generation. Each task is assigned its own target return, and in our experiments we directly adopt the corresponding hyperparameters from the cleandiffuser reimplementation of Decision Diffuser.

### B.2.2 OFFLINE RL ALGORITHM IMPLEMENTATION

In our experiments, we evaluate BiTrajDiff against four offline RL algorithms: IQL, TD3BC, CQL, and DT. To enable efficient and large-scale evaluation, we conduct all offline RL experiments using the JAX-based library JAX-CORL (Nishimori, 2024). **For fairness, we report re-run results on JAX-CORL with initial datasets as baseline performance, rather than the scores reported in the original papers, to account for discrepancies across codebases.** Meanwhile, for each offline RL algorithm, we ensure that the hyperparameters are configured consistently with those reported in the respective original papers.

## C EXPERIMENTAL RESULTS ON CQL AND DT

**Results for Locomotion Tasks.** The results on D4RL locomotion tasks are shown in Table S2. BiTrajDiff achieves the highest average returns in all of the environments under both CQL and DT baselines, with improvements of 5.6% and 7.4%, respectively. At the same time, BiTrajDiff maintains high stability across all locomotion tasks, substantially exceeding other tested DA methods. These results highlight the effectiveness of BiTrajDiff.

**Results for Maze and Franka Kitchen Tasks.** Table S3 shows the experimental results on the D4RL Maze and Franka Kitchen tasks. BiTrajDiff acquires the highest average return across all tested tasks of sparse reward scenarios under both CQL and DT baselines. Meanwhile, BiTrajDiff maintains higher stability in performance improvements across all the settings than other DA methods like RTDiff. This observation confirms that BiTrajDiff can synthesize novel and high-quality long-horizon trajectories by stitching forward-future and backward-history paths to address the challenge of the availability of high-quality data under sparse-reward settings. This result underlines the efficacy and applicability of BiTrajDiff in enhancing data diversity and quality.

Table S3: Test returns of our proposed BiTrajDiff and baselines on Maze and Franka Kitchen tasks.

| Task | CQL (Kumar et al., 2020) | | | | | DT (Chen et al., 2021) | | | | |
|---|---|---|---|---|---|---|---|---|---|---|
| | Base | RTDiff | Synther | DiffStitch | Ours | Base | RTDiff | Synther | DiffStitch | Ours |
| maze2d-umaze | 4.5±1.1 | -11.3±8.3 | 2.5±4.6 | -2.3±1.4 | **18.6**±29.4 | 24.9±10.6 | 29.8±14.5 | 22.9±2.7 | **36.1**±8.7 | **36.1**±8.7 |
| maze2d-medium | 84.4±24.5 | 87.7±23.2 | 90.5±10.7 | 72.9±16.8 | **95.8**±17.4 | 16.6±2.2 | 18.8±2.7 | 19.3±1.6 | **26.6**±4.0 | **26.6**±4.0 |
| maze2d-large | 34.3±25.5 | 38.9±20.7 | 45.3±14.2 | **65.2**±72.5 | 48.3±28.7 | 22.3±12.5 | 24.9±4.9 | 24.4±7.3 | **28.5**±4.4 | **28.5**±4.4 |
| antmaze-umaze-diverse | 31.6±8.4 | 40.6±9.0 | 42.4±6.8 | 39.2±9.3 | **48.8**±4.4 | 42.0±4.9 | 46.4±16.7 | 42.6±4.7 | **47.8**±2.9 | **47.8**±2.9 |
| antmaze-medium-diverse | 57.2±16.2 | 60.4±12.8 | 62.6±9.8 | 63.6±12.6 | **73.2**±11.6 | 0.0±0.0 | 0.2±0.4 | 0.0±0.0 | **0.4**±0.5 | **0.4**±0.5 |
| antmaze-large-diverse | 8.0±7.3 | 5.3±1.2 | 12.6±10.9 | 2.6±3.7 | **16.0**±10.2 | 0.0±0.0 | 0.0±0.0 | 0.0±0.7 | **0.8**±0.7 | **0.8**±0.7 |
| **Maze Mean** | 36.7 | 38.6 | 42.4 | 37.6 | **48.8** | 17.5 | 19.3 | 18.2 | 22.3 | **22.6** |
| kitchen-complete | 7.8±11.9 | 15.2±9.3 | 8.4±6.1 | 10.5±5.3 | **13.8**±7.4 | 61.7±14.6 | 61.3±8.0 | 63.4±11.7 | **67.9**±6.8 | **67.9**±6.8 |
| kitchen-partial | 21.1±2.5 | 25.8±3.1 | 23.2±7.3 | 19.3±6.7 | **24.9**±4.7 | 21.6±11.1 | 22.2±7.5 | 29.2±5.8 | **33.5**±18.9 | **33.5**±18.9 |
| kitchen-mixed | 16.2±6.4 | 19.1±1.2 | 19.3±5.7 | 12.0±9.6 | **21.5**±7.9 | 22.5±16.5 | 23.7±14.7 | 24.7±10.4 | **37.0**±11.7 | **37.0**±11.7 |
| **Kitchen Mean** | 15.0 | 20.0 | 17.0 | 13.9 | **20.1** | 35.3 | 35.7 | 39.1 | 46.1 | **46.1** |

Table S4: Test returns of our proposed BiTrajDiff with augmented single- and bi-directional diffusion generated trajectories on IQL and TD3BC algorithms under D4RL locomotion tasks.

| Source | Task | IQL (Kostrikov et al., 2021) | | | | TD3BC (Fujimoto and Gu, 2021) | | | |
|---|---|---|---|---|---|---|---|---|---|
| | | Base | Forward | Backward | Ours | Base | Forward | Backward | Ours |
| medium | halfcheetah | 48.2±0.2 | 47.3±0.8 | 48.4±0.5 | **48.6**±0.2 | 48.4±0.2 | 50.0±0.3 | **50.8**±0.7 | 50.3±0.4 |
| | walker2d | 77.6±5.3 | 79.1±4.9 | 83.7±2.6 | **86.7**±1.7 | 82.0±2.7 | 81.7±0.7 | 84.6±0.4 | **86.7**±1.0 |
| | hopper | 67.0±1.9 | 70.2±7.6 | 75.8±4.8 | **81.1**±5.0 | 60.4±3.7 | 65.2±6.7 | **80.3**±4.8 | 79.0±3.5 |
| medium expert | halfcheetah | 90.1±4.4 | 92.5±1.2 | 94.2±0.1 | **95.3**±0.1 | 92.3±5.1 | 94.7±1.9 | 95.3±0.9 | **96.3**±0.4 |
| | walker2d | 112.4±0.4 | 109.7±1.6 | **112.5**±0.4 | **112.5**±0.5 | 109.3±0.2 | 109.1±0.5 | 110.0±0.4 | **110.2**±0.3 |
| | hopper | 105.4±6.3 | 102.3±7.0 | 109.1±5.2 | **110.9**±0.6 | 94.2±11.1 | 88.7±5.5 | 108.4±2.5 | **109.0**±5.1 |
| medium replay | halfcheetah | 43.5±0.2 | 42.5±0.8 | 43.7±0.1 | **44.1**±0.2 | 44.1±0.1 | 44.4±0.3 | **45.1**±0.5 | 45.0±0.6 |
| | walker2d | 73.3±7.5 | 79.4±5.0 | 82.2±2.9 | **87.6**±2.2 | 80.9±6.0 | 87.2±2.4 | 85.8±4.9 | **89.9**±1.4 |
| | hopper | 94.7±4.0 | 97.2±3.0 | 89.6±3.1 | **102.8**±0.6 | 64.4±8.9 | 83.5±12.9 | 83.8±9.7 | **85.0**±10.3 |
| **Average** | | 79.1 | 78.9 | 82.1 | **85.5** | 75.1 | 78.3 | 82.7 | **85.7** |

Table S5: Test returns of our proposed BiTrajDiff with augmented single- and bi-directional diffusion generated trajectories on CQL and DT algorithms under D4RL locomotion tasks.

| Source | Task | CQL (Kumar et al., 2020) | | | | DT (Chen et al., 2021) | | | |
|---|---|---|---|---|---|---|---|---|---|
| | | Base | Forward | Backward | Ours | Base | Forward | Backward | Ours |
| medium | halfcheetah | 47.4±0.2 | 47.5±0.2 | **48.9**±0.4 | 48.7±0.3 | 42.7±0.5 | 42.7±0.2 | 43.0±0.3 | **43.3**±0.2 |
| | walker2d | 82.6±1.0 | 80.9±2.7 | 83.8±0.7 | **84.2**±0.5 | 69.6±11.0 | 74.6±3.6 | 76.1±3.5 | **77.8**±4.0 |
| | hopper | 68.8±4.1 | 71.6±14.9 | 75.2±9.5 | **80.9**±3.4 | 56.0±1.9 | 56.6±5.6 | 56.9±5.7 | **57.8**±2.2 |
| medium expert | halfcheetah | 87.2±4.6 | 88.0±3.9 | 92.6±2.2 | **94.1**±1.1 | 70.8±12.3 | 74.3±4.7 | 64.7±4.4 | **83.5**±6.3 |
| | walker2d | 110.2±0.6 | **110.5**±0.6 | 109.5±0.7 | 110.4±0.4 | 104.6±6.9 | 105.1±5.1 | 104.9±4.8 | **106.3**±6.0 |
| | hopper | 103.5±8.1 | 102.7±13.9 | 102.5±7.8 | **105.7**±5.8 | 107.5±3.4 | 108.3±3.3 | 109.8±2.4 | **110.1**±1.9 |
| medium replay | halfcheetah | 45.9±0.2 | 45.9±0.5 | **46.2**±0.4 | 45.8±0.3 | 38.8±2.4 | 36.4±2.9 | 39.3±0.5 | **39.5**±0.8 |
| | walker2d | 82.1±1.8 | 82.9±0.6 | 84.4±2.1 | **84.8**±1.5 | 52.1±5.5 | **58.6**±10.0 | 54.9±4.2 | 58.2±6.1 |
| | hopper | 95.7±2.5 | 97.9±6.1 | 93.8±5.3 | **100.3**±1.3 | 67.3±8.9 | 62.1±13.0 | 68.4±17.2 | **69.6**±13.1 |
| **Average** | | 80.4 | 80.9 | 81.9 | **83.9** | 67.7 | 68.7 | 68.7 | **71.8** |

# D EXPERIMENTAL RESULTS WITH THE DIRECTION OF DIFFUSION GENERATION

We conduct experiments with different augmented diffusion trajectories generated in single-directional and bi-directional. Table S4 shows the test returns of the IQL and TD3BC algorithms. For CQL and DT algorithms, the quantitative results are reported in Table S5. The algorithms were evaluated on D4RL locomotion tasks with augmented trajectories generated in single-direction and bi-directional.

Table S6: Test results of BiTrajDiff with data from different variants related to trajectory filters. OOD indicates the OOD filter, and G represents the greedy filter.

| Source | Task | IQL (Kostrikov et al., 2021) | | | | | TD3BC (Fujimoto and Gu, 2021) | | | | |
|---|---|---|---|---|---|---|---|---|---|---|---|
| | | Base | None | w G, w/o OOD | w OOD, w/o G | Ours | Base | None | w G, w/o OOD | w OOD, w/o G | Ours |
| medium replay | walker2d | 73.3±7.5 | 78.5±10.2 | 83.9±4.5 | 80.1±7.0 | **87.6**±2.2 | 80.9±6.0 | 77.2±24.4 | 77.5±26.9 | 82.5±8.0 | **89.9**±1.4 |
| | hopper | 94.7±4.0 | 93.6±12.0 | 91.6±8.2 | 96.3±3.5 | **102.8**±0.6 | 64.4±8.9 | 69.0±19.2 | 75.0±17.0 | 74.6±23.6 | **85.0**±10.3 |

Compared to the base method, the forward method offers only modest gains and is at times even detrimental to performance. Conversely, the backward method generally outperforms the base method across most tasks, achieving substantial gains. Finally, our bi-directional approach, BiTrajDiff, consistently exhibits superior and robust performance. Our BiTrajDiff enables global connectivity between originally unreachable states from the original dataset and thus constructs novel and high-quality augmented trajectories.

## E    EXPERIMENTAL RESULTS WITH TRAJECTORY FILTERS

We illustrate the necessity of integrating both OOD and greedy filters in our full BiTrajDiff model. While Figure 3 presents the learning curves, Table S6 directly compares the performance on walker2d-medium-replay and hopper-medium-replay tasks under different filter settings. Without any filters, the performance is comparable to, and in some cases inferior to, that of the base method on both the IQL and TD3BC algorithms. When applying the greedy filter in isolation, the performance improvement is noticeable but marred by volatility. In contrast, using the OOD filter alone yields a more stable and substantial performance enhancement. Nevertheless, our full BiTrajDiff model, which integrates both filters, consistently outperforms all other configurations across the tested settings, indicating the necessity of both OOD and greedy filters.

## F    EXPERIMENTAL RESULTS WITH AUGMENTED DATA RATIO $\sigma$

In this section, we conduct experiments to assess the impact of the augmented data ratio $\sigma$. As shown in Table S1, when the ratio $\sigma$ is too low, which is between $0\% \sim 20\%$, performance improvements remain consistently constrained. Conversely, while an augmentation ratio $\sigma$ between $30\%$ and $50\%$ already provides a substantial and stable performance improvement, increasing $\sigma$ to $100\%$ compromises this stability, indicating a decline in the performance. This result demonstrates that both the limited and overabundant synthetic data have detrimental effects on the performance of BiTrajDiff, since the limited data is not enough to provide offline RL algorithms with new behavior patterns, and overabundant data introduces excessive noise, which hinders offline RL algorithms from learning effectively, and an optimal trade-off is achieved with a $30\%$ to $50\%$ augmentation range. As a result, we choose $\sigma = 30\%$ as the fixed hyperparameter for all our experiments.

Figure S1: Performance comparison of Bi-TrajDiff with different augmented data ratios $\sigma$ in the walker2d-medium-replay task. The best result for each offline RL algorithm is marked as **bold**.

| Ratio | IQL | TD3BC | CQL |
|---|---|---|---|
| 0% | 73.3±7.5 | 80.9±6.0 | 82.1±1.8 |
| 10% | 78.7±2.4 | 82.0±5.5 | 84.0±3.1 |
| 20% | 84.2±1.3 | 87.5±1.0 | 83.9±2.0 |
| 30% | 87.6±2.2 | **89.9**±1.4 | 84.8±1.5 |
| 40% | **88.7**±4.5 | 86.7±2.9 | 84.7±2.1 |
| 50% | 81.6±5.6 | 86.4±2.7 | **86.3**±2.1 |
| 100% | 84.5±10.2 | 86.7±2.4 | 83.9±3.3 |

## G    EXPERIMENTAL RESULTS WITH $n$-STEP TD ESTIMATOR

In this section, we demonstrate the effectiveness of BiTrajDiff in Offline RL algorithms with varying lengths of the $n$-step TD estimators. As shown in Table S7 and Table S8, while RTDiff and diffsitch initially outperform the base method, their performances exhibit a sharp decline, even being surpassed by the base method, as $n$ increases. Conversely, our proposed BiTrajDiff generally achieves the highest scores and consistently maintains its advantage as the $n$-step horizon increases. Meanwhile,

Table S7: Test returns of our proposed BiTrajDiff and baselines with varying length of $n$-step TD estimator ($n$) on walker2d-medium-replay task.

| Length of n-step TD Estimator($n$) | IQL (Kostrikov et al., 2021) | | | | TD3BC (Fujimoto and Gu, 2021) | | | |
|---|---|---|---|---|---|---|---|---|
| | base | RTDiff | diff stitch | ours | base | RTDiff | diff stitch | ours |
| $n=2$ | $73.3_{\pm7.5}$ | $74.0_{\pm5.2}$ | $\underline{74.9}_{\pm8.2}$ | $\mathbf{87.6}_{\pm2.2}$ | $80.9_{\pm6.0}$ | $\underline{83.5}_{\pm7.4}$ | $80.5_{\pm5.2}$ | $\mathbf{89.9}_{\pm1.4}$ |
| $n=5$ | $71.7_{\pm12.4}$ | $76.4_{\pm16.6}$ | $\underline{79.8}_{\pm2.4}$ | $\mathbf{86.7}_{\pm0.9}$ | $80.3_{\pm12.0}$ | $\mathbf{88.6}_{\pm3.6}$ | $\underline{87.7}_{\pm4.0}$ | $87.3_{\pm1.3}$ |
| $n=10$ | $69.5_{\pm13.5}$ | $\underline{73.1}_{\pm12.7}$ | $71.4_{\pm9.9}$ | $\mathbf{79.5}_{\pm2.3}$ | $\underline{81.5}_{\pm7.6}$ | $79.6_{\pm12.7}$ | $81.3_{\pm5.1}$ | $\mathbf{87.5}_{\pm3.4}$ |
| $n=15$ | $\underline{72.7}_{\pm11.4}$ | $67.2_{\pm14.8}$ | $69.7_{\pm6.0}$ | $\mathbf{77.9}_{\pm6.2}$ | $75.1_{\pm9.9}$ | $\underline{80.6}_{\pm13.8}$ | $79.1_{\pm9.7}$ | $\mathbf{83.9}_{\pm4.7}$ |

Table S8: Test returns of our proposed BiTrajDiff and baselines with varying length of $n$-step TD estimator ($n$) on hopper-medium-replay task.

| Length of n-step TD Estimator($n$) | IQL (Kostrikov et al., 2021) | | | | TD3BC (Fujimoto and Gu, 2021) | | | |
|---|---|---|---|---|---|---|---|---|
| | base | RTDiff | diff stitch | ours | base | RTDiff | diff stitch | ours |
| $n=2$ | $94.7_{\pm4.0}$ | $\underline{97.7}_{\pm2.7}$ | $96.4_{\pm5.6}$ | $\mathbf{102.8}_{\pm0.6}$ | $64.4_{\pm8.9}$ | $\underline{72.1}_{\pm18.9}$ | $71.5_{\pm21.0}$ | $\mathbf{85.0}_{\pm10.3}$ |
| $n=5$ | $90.5_{\pm1.9}$ | $94.7_{\pm5.4}$ | $\mathbf{99.9}_{\pm6.3}$ | $\underline{98.7}_{\pm6.4}$ | $67.0_{\pm96.6}$ | $\underline{75.3}_{\pm14.8}$ | $67.7_{\pm21.0}$ | $\mathbf{89.7}_{\pm12.6}$ |
| $n=10$ | $94.5_{\pm9.8}$ | $\underline{97.0}_{\pm2.0}$ | $93.3_{\pm9.1}$ | $\mathbf{100.0}_{\pm1.3}$ | $62.7_{\pm25.5}$ | $\underline{66.6}_{\pm15.6}$ | $63.3_{\pm25.7}$ | $\mathbf{83.5}_{\pm13.5}$ |
| $n=15$ | $82.0_{\pm26.9}$ | $\underline{93.3}_{\pm9.1}$ | $88.4_{\pm9.5}$ | $\mathbf{99.5}_{\pm6.8}$ | $\underline{65.0}_{\pm26.0}$ | $64.8_{\pm25.7}$ | $39.6_{\pm17.2}$ | $\mathbf{71.3}_{\pm14.3}$ |

as described in section 5.4, the performance of all tested methods degrades with an increasing $n$-step horizon, which is attributed to the escalating variance in value estimation. Moreover, for DA methods, the accumulated errors introduced by the inverse dynamics model should also be considered. In contrast, BiTrajDiff alleviates these issues by synthesizing novel, high-quality long-horizon trajectories, thereby achieving a more stable and consistent performance. This result demonstrates that by stitching forward-future and backward-history trajectories, BiTrajDiff synthesizes higher-quality and more reliable trajectories than existing DA methods.

# H  THE USE OF LARGE LANGUAGE MODELS (LLMS)

In this work, we leverage Large Language Models (LLMs) to enhance and refine textual content. Specifically, LLMs are employed to polish paragraphs, improving their fluency, coherence, grammatical accuracy, and overall readability.

