# OpenReview forum: "BiTrajDiff: Bidirectional Trajectory Generation with Diffusion Models for Offline Reinforcement Learning"
_ICLR.cc/2026/Conference — ICLR 2026 Conference Withdrawn Submission_

### Official Review · Reviewer_QVeW · 2025-10-24

**Soundness:** 2
**Presentation:** 2
**Contribution:** 2
**Rating:** 2
**Confidence:** 4

**Summary:**

This paper proposes a diffusion-based method that, given a specific state, generates both the subsequent trajectory leading from this state and the historical trajectory leading to it. Although BiTrajDiff achieves relatively good performance in the experiments, the reasons behind this improvement are not clearly explained.

**Strengths:**

BiTrajDiff demonstrates consistently good results across multiple environments.

**Weaknesses:**

1. Main weakness: The motivation of this work is unclear. Specifically, it does not explain why it is necessary to generate both historical and future trajectories simultaneously, nor what advantages such joint generation provides.

2. Lack of novelty: The proposed method appears quite similar to DiffStitch, and its distinct innovation is not evident.

3. Clarity issues: The overall organization of the paper lacks clarity. For instance, Section 4.1 mixes the processes of generating future and historical trajectories, making it difficult for readers with limited background knowledge to follow.

4. Misstatements: The authors claim that prior works (such as DiffStitch) only generate future trajectories based on the current state and cannot generate historical ones. However, DiffStitch actually generates intermediate states between a low-return state and a high-return state from the dataset. Therefore:
(1) DiffStitch is not limited to generating future trajectories from the current state;
(2) From the perspective of the high-return state, DiffStitch is effectively generating historical trajectories.

**Questions:**

1. Why is it necessary to generate both historical and future trajectories? What are the advantages of doing so?

2. What are the key design differences between BiTrajDiff and DiffStitch?

3. In Section 5.4, how are the rewards of the generated trajectories evaluated?

4. It would be helpful if the authors could clarify how the generated data are ensured to go beyond the offline dataset. In Section 4.2.3, samples are filtered based on distance, whereas in Section 5.5 the L2 distance between generated and dataset trajectories appears to be limited. These two descriptions seem somewhat inconsistent and warrant further explanation.

---

### Official Review · Reviewer_Rfoc · 2025-10-30

**Soundness:** 1
**Presentation:** 2
**Contribution:** 1
**Rating:** 2
**Confidence:** 5

**Summary:**

The paper proposes a novel data augmentation algorithm for offline reinforcement learning RL, BiTrajDiff, which utilizes the diffusion model with conditional generation. The central claim of the authors is that generating both future and historical trajectories is a key recipe for data augmentation. In the experiment section, BiTrajDiff outperforms other diffusion-based data augmentation baselines.

**Strengths:**

1. BiTrajDiff Improves performance of offline RL baselines

The main results show that augmenting offline RL baselines with BiTrajDiff improves performance. Moreover, BiTrajDiff quantitatively shows that it expands diversity while maintaining accuracy, enabling larger performance gains.

**Weaknesses:**

1. Critically overlapped with the GTA [1]

BiTrajDiff generates trajectories using a diffusion model trained on sub-trajectories of an offline dataset, with classifier-free guidance conditioned on discounted returns. GTA augments the dataset with a trajectory-level diffusion model that samples high-quality data conditioned on the amplified discounted return of the initial state. Both approaches share three critical components: (1) trajectory-level diffusion training, (2) classifier-free guided conditional generation, and (3) return-conditioned trajectory augmentation. Although the two methods are very similar, and GTA only augments future trajectories, BiTrajDiff underperforms GTA on the D4RL gym locomotion benchmark (e.g., IQL-gym locomotion: 85.5 vs. 86.11; TD3BC-gym locomotion: 83.5 vs. 84.63). Thus, given the methodological overlap and inferior performance, BiTrajDiff’s contribution to the offline RL literature seems to be limited.

2. Complicated method with five trainable components

BiTrajDiff comprises five learnable components: two parameterized diffusion models $\theta^b, ~\theta^k$, inverse dynamics model $f_\phi$, reward model $r_\phi$, OOD trajectory filter with isolation forest. Each of them requires its own hyperparameters, necessitating an extensive search through the combinatorial space of the learnable components. Furthermore, the OOD trajectory filter and greedy trajectory filter require additional hyperparameters Top-$C_\text{ood}$ and Top-$C_\text{greedy}$. Conditional generation requires target return to go, $R^f$, and $R^b$. One of the most important virtues of the offline RL is simplicity [2]. However, BiTrajDiff is a combination of five models with additional hyperparameters, which hinders its implication for real-world applications.

3. Limited reproducibility due to missing hyperparameter settings and implementation details

The paper does not provide any hyperparameter configurations or implementation details for the inverse dynamics model, reward model, OOD trajectory filter, and greedy trajectory filter.

[1] Lee, Jaewoo, et al. "Gta: Generative trajectory augmentation with guidance for offline reinforcement learning." Advances in Neural Information Processing Systems 37 (2024): 56766-56801.

[2] Fujimoto, Scott, and Shixiang Shane Gu. "A minimalist approach to offline reinforcement learning." Advances in neural information processing systems 34 (2021): 20132-20145.

**Questions:**

1. Justification for omitting GTA as a baseline.

Except for the generation direction, BiTrajDiff and GTA are largely similar. What is the reasonable justification for excluding GTA from the baselines? Differences in implementation frameworks are not an issue, because the experimental scope matches exactly. It is sufficient to report the published results as-is.

2. Setting of $R^f$ and $R^b$

The paper does not clearly specify how to set these two hyperparameters. GTA reports that the choice of conditioning value has a substantial impact on performance. What is the corresponding impact in BiTrajDiff?

---

### Official Review · Reviewer_jBXs · 2025-10-30

**Soundness:** 2
**Presentation:** 3
**Contribution:** 1
**Rating:** 2
**Confidence:** 3

**Summary:**

This paper proposes BiTrajDiff, a bidirectional diffusion-based data augmentation framework for offline reinforcement learning. Unlike existing methods that generate only forward trajectories from initial states, BiTrajDiff introduces a bidirectional trajectory generation paradigm that simultaneously synthesizes both forward-future and backward-history trajectories conditioned on shared intermediate “anchor” states. While the approach shows promising results across various environments and demonstrates advantages in handling sub-optimal datasets, several limitations remain, such as lacking of in-depth analysis, insufficiently comprehensive ablation study.

**Strengths:**

1. The paper is well-structured and easy to follow, and the proposed method is thoroughly validated on sufficiently large datasets.
2. The experiments are comprehensive, especially the ablation study on the n-step TD estimator. In addition, the authors provide detailed implementation and hyperparameter information, which facilitates reproducibility of the method.
3. The method is well-designed, incorporating novel filtering mechanisms (including an OOD trajectory filter and a greedy trajectory filter) to exclude out-of-distribution (OOD) and suboptimal trajectories, thereby ensuring the high quality and reliability of the augmented dataset.

**Weaknesses:**

1. The motivation is questionable. The authors claim that by generating “forward-future” and “backward-history” trajectories from anchor states, the method can “connect” previously unreachable state pairs in the dataset and thus improve data diversity. However, as shown in Algorithm 1, the anchor states are entirely sampled from the existing offline dataset D. Therefore, the generated trajectories still fluctuate around the high-density regions of the original data distribution around the anchor states, which does not truly go beyond the existing dataset.

2. The experimental validation is insufficient. As mentioned in Weaknesses 2, the method appears equivalent to reproducing the dataset distribution and sampling from it unconditionally. However, the ablation study does not include any analysis to verify this assumption.

3. The paper lacks analysis and validation for several key hyperparameters, such as $C_{ood}$ and $C_{greedy}$, as well as $R^f$ and $R^b$.

4. Unclear description of technical details. For instance, in Algorithm 1, does the dataset size |D| refer to the number of trajectories or the number of  $(s_t, a_t, s_{t+1})$ transitions?

5. There is some controversy regarding the scale used in Figure 6, as it reports absolute errors. Since the range of state values varies across different environments, using relative errors would generally provide a more meaningful measure of the error magnitude.

**Questions:**

1. As is shown in Algorithm 1, the anchor states are sampled uniformally from the dataset. How does BiTrajDiff ensure that the generated trajectories truly expand beyond the original distribution rather than merely reproducing its local statistics?
2. In Algorithm 1, does the dataset size |D| refer to the number of trajectories or the number of  $(s_t, a_t, s_{t+1})$ transitions?
3. In Figure 2, $\textit{forward }$ data augmentation can lead to negative effects in some cases (eg, h-m-r of IQL, h-m-e of TD3BC, h-m-r of DT and hc-m-e of DT). Could the authors provide a detailed explanation?
4. In Appendix B.2.1, it is mentioned that the default horizon for generated trajectories is 5. Is this too short relative to the full trajectory length (which can typically reach 1000 in MuJoCo environments)? Could the authors conduct experiments regarding the choice of horizon for trajectory generation?

---

### Official Review · Reviewer_YTkw · 2025-10-31

**Soundness:** 3
**Presentation:** 3
**Contribution:** 3
**Rating:** 4
**Confidence:** 3

**Summary:**

This paper introduces BiTrajDiff, modeling both forward-future and backward-history trajectories from intermediate state, enabling the reconstruction of globally diverse behavioral patterns beyond those in the dataset. To ensure consistency and realism, BiTrajDiff integrates an Inverse Dynamics Model (IDM) and a Reward Model (RM) to infer actions and rewards, followed by a two-stage filtering mechanism — an OOD filter (via Isolation Forest) and a greedy filter that selects high-return samples. Empirical evaluation on the D4RL benchmark demonstrates that BiTrajDiff consistently outperforms strong data augmentation baselines across multiple offline RL algorithms, including IQL, TD3+BC, CQL, and Decision Transformer (DT).
- An LLM was used to improve writing.

**Strengths:**

1. Novel bidirectional design: The paper introduces a conceptually novel idea of using two diffusion model to jointly model forward and backward trajectories.

2. Robust filtering and completion pipeline: The combination of an IDM, RM, and dual-stage filtering (OOD + greedy) ensures the augmented data maintains both plausibility and reward relevance.

3. Strong empirical results: Extensive experiments on D4RL locomotion, navigation, and manipulation tasks show consistent and significant gains, with well-designed ablations confirming the contribution of each component.

**Weaknesses:**

1. Limited qualitative analysis of generated trajectories.
While the paper presents quantitative results demonstrating performance improvements, it would be helpful to include qualitative examples showing how the generated trajectories differ from the originals.
For instance, given a selected anchor state, what was the original trajectory it came from, and how did the augmented (bidirectional) trajectory change its structure or behavior? Such visual or descriptive comparisons would provide stronger intuition about how BiTrajDiff actually enhances trajectory diversity and realism.

2. Missing evaluation of oracle rewards and baseline comparison for trajectory quality metrics.
The paper reports L2 distance and dynamic error for BiTrajDiff in Figure 6, but does not include comparisons with baseline augmentation methods (e.g., Synther, RTDiff, DiffStitch).
It would strengthen the empirical section to show how BiTrajDiff performs relative to these baselines on the same metrics.
Additionally, reporting oracle reward statistics for the generated trajectories would provide a more direct assessment of their quality and potential utility for policy improvement.
An ablation showing how these metrics change without the filtering would also clarify the importance of the filtering step.

3. Dependence on anchor states: The method’s success partly hinges on how intermediate “anchor” states are selected, but this process is sampled randomly without clear adaptive selection.


4. Incomplete coverage of related work on trajectory-level augmentation.
The related work section overlooks several recent studies that are conceptually or methodologically relevant to BiTrajDiff’s contributions. For instance, [1] apply simple transformations such as noise injection to pixel-based or state-based observation, [2] leverages diffusion model to generate synthetic trajectory for online RL, and [3] introduces guided trajectory augmentation with diffusion models.
Including and discussing these works would help position BiTrajDiff more clearly within the broader landscape of trajectory generation and data augmentation for offline RL. It may also be worth clarifying whether these approaches could serve as additional baselines for comparison in future revisions.




[1] S4RL: Surprisingly Simple Self-Supervision for Offline Reinforcement Learning in Robotics
[2] ATraDiff: Accelerating Online Reinforcement Learning with Imaginary Trajectories
[3] GTA: Generative Trajectory Augmentation with Guidance for Offline Reinforcement Learning

**Questions:**

See weaknesses.

---

### Note · Authors · 2025-11-28

**Comment:**

We would like to formally withdraw our paper from the ICLR 2026 submission process. After careful consideration, we have decided to revise and enhance our work based on the constructive feedback we have received. We believe these revisions will strengthen our contribution and make it more impactful. We are grateful for the time and thoughtful feedback provided by the reviewers and Area Chairs throughout this process.

**Withdrawal Confirmation:**

I have read and agree with the venue's withdrawal policy on behalf of myself and my co-authors.